## RESEARCH ARTICLE

# Depression, anxiety and change in eating habits during the COVID-19 pandemic in Brazilian university students

Marcus Verly-Miguel[1]*, Claudia de Souza Lopes[1], Jade Veloso Freitas[1‡], Magno Conceição Garcia[1‡], Marcio Candeias Marques[1‡], Vitor Barreto Paravidino[1,2,3], Rosely Sichieri[1]

1 Department of Epidemiology, Social Medicine Institute, State University of Rio de Janeiro, Rio de Janeiro, Brazil, 2 National School of Public Health, Oswaldo Cruz Foundation, Rio de Janeiro, Brazil, 3 Department of Physical Education and Sports, Naval Academy-Brazilian Navy, Rio de Janeiro, Brazil

☯ These authors contributed equally to this work.
‡ JVF and MCG also contributed equally to this work.
* mmiguel@id.uff.br

## Abstract

This cross-sectional study aims to evaluate the association between anxiety and depression with changes in the consumption of hyperpalatable foods and meal patterns in a sample of 771 Brazilian university students during the social isolation period in the COVID-19 pandemic. More than half of the subjects self-reported clinically significant symptoms of anxiety (53.8%) and depression (62.5%), with 47.6% having both. Most individuals who showed increased consumption of hyperpalatable foods were also part of the group that reported clinically significant symptoms of anxiety or depression. Statistical analysis was performed using exploratory structural equations. The latent variable "symptoms of anxiety and depression" was created using the anxiety and depression scores. Symptoms of anxiety and depression had a positive correlation with the increased consumption of hyperpalatable foods and meal substitution (standardized coefficient = 0.212), after analysing their total direct and indirect effects. It was concluded that higher scores of anxiety and depression negatively affects the eating habits of university students.

## Introduction

Social isolation measures began to be implemented in several Brazilian states from the second week of March 2020 [1], three weeks after the first recorded case of COVID-19 in Brazil [2]. These measures included the closure of schools and universities, non-essential businesses and public leisure areas as well as the adoption of remote work (home office), among others. In general, they were accepted by a large part of the population [3]. Although these measures were crucial in controlling the spread of the virus, social isolation had negative impacts on the population, such

**Data availability statement:** The database, syntax and questionnaire files are available from the figshare database. Direct link: https://figshare.com/s/153b348888e5fde6de24 DOI: 10.6084/m9.figshare.25750836

**Funding:** Funding received according to the grant n° 255937: Emergencial Action COVID-19 - Call C - Support for Research Network Projects on SARS-CoV-2/COVID-19. Funding provided by Carlos Chagas Filho Foundation (FAPERJ). The funders had no role in study design, data collection and analysis, decision to publish, or preparation of the manuscrip.

**Competing interests:** The authors have declared that no competing interests exist.

as changes in eating behaviour and weight gain [4,5], as well as an increase in the prevalence of depression and anxiety [6].

Fat, sugar, sodium and simple carbohydrates, present in moderate to high levels in foods, can synergistically and artificially increase their palatability [7]. These hyperpalatable are positively associated with increased weight gain in adults [8], and development of obesity [9,10]. In addition, hyperpalatable foods can lead to behavioural and biological changes associated with food addiction [7,8,11,12]. Moreover, hyperpalatable foods and snacks account for approximately 70% of food choices on Brazilian food delivery apps [13], an industry that experienced a surge in popularity during the pandemic [14].

The social isolation provided a unique opportunity to examine the relationship between anxiety and depression, factors that may negatively impacts eating habits [15] and are known to affect males and females differently [16]. A high prevalence of anxiety and depression disorders was found in two cross-sectional studies conducted in the Brazilian university population during the social isolation [17,18] however, no study to date has evaluated the association between anxiety and depression with eating habits. Therefore, the present study aims to contribute to the understanding of possible changes in food intake during the social isolation period and whether these changes are associated with depression and anxiety, based on the hypothesis that higher levels of these symptoms are linked to increased consumption of hyperpalatable foods.

## Methods

### Population and study design

This is a cross-sectional study of students who enrolled in undergraduate degree programs of a public university in the state of Rio de Janeiro in 2019, and whose data were collected by sending a questionnaire and an informed consent form by email.

A total of 4,935 students enrolled at the university in 2019, across 32 undergraduate courses. Of these, only 3,973 (80,5%) had valid email addresses and 771 (15,6%) students answered the questionnaire.

The sampling power to capture differences in relation to population size was 0.8, the sample size using a proportion of 50% and an error of 0.05 was 384.

The data was collected between August 6, 2020 and March 13, 2021, a period where there were no available COVID-19 vaccines in Brazil.

### Sample weighting

Due to the high non-response rate (80.6%), analyses were weighted based on total frequency by sex and course of the students who entered the university. The proportion of each sex and the number of individuals in each course was obtained from the university's statistical yearbook [19].

For example, the total number of students entering the administration course in 2019 was 103 individuals, being 47 females and 56 males. The number of respondents for the same course was 11 individuals, being 8 females and 3 males. Thus, a weight of 5.9 was assigned to each female individual (47/8 = 5.9) and 18.7 to each male individual (56/3 = 18.7).

## Questionnaire

A Google Form questionnaire was developed based on validated instruments commonly used in epidemiological studies [20,21]. It included 29 food-items frequently reported in dietary surveys in Brazil [22], as along with basic respondent information, such as e-mail address. For each food item, participants were asked whether their consumption had increased, decreased, or remained unchanged during the pandemic. Changes in eating habits were assessed through three specific questions: 1) "During the pandemic how did the frequency of consumption of ready-to-eat foods via delivery change?; 2) How did your consumption of snacks at dinner (e.g., sandwiches, pizzas, snacks, snacks, hamburgers) change?; and 3) How did your consumption of snacks at lunch (e.g., sandwiches, pizzas, snacks, snacks, hamburgers) change?".

Anxiety symptoms were measured by the Brazilian version of the 7-item Generalized Anxiety Disorder Scale (GAD-7), which investigates the presence of self-reported anxiety symptoms in the last two weeks [21,23]. Each item on the scale can receive a score ranging from 0 to 3, where "never" receives 0, "some days" receives 1, "more than half of the days" receives 2 and "almost every day" receives 3. The maximum score achieved on the scale is 21 points. The cut-off points chosen were "no symptoms" for those with scores ≤4, "mild" for those with scores between 5 and 9, and "moderate" for those with scores between 10 and 14, and "severe" for those with scores ≥15. The cut-off points were defined based on the literature on the scale [23].

Depression symptoms were measured using the Brazilian version of the Patient Health Questionnaire 9 (PHQ-9), a scale that investigates the presence of self-reported depression symptoms in the last two weeks [20,24]. Each item on the scale can receive a score ranging from 0 to 3, where "never" receives 0, "some days" receives 1, "more than half of the days" receives 2 and "almost every day" receives 3. The maximum score achieved on the scale is 27 points. The cut-off points chosen were "no symptoms " for those with scores ≤4, "mild" for those with scores between 5–9, "moderate" for those with scores between 10–14, "moderately severe" for those with scores between 15–19, and "severe" for those with scores ≥20. The cut-off points were defined based on the literature on the scale [25].

## Study variables

Beyond the scores of GAD-7 and PHQ-9, age, sex, skin colour, marital status, income variation, time of isolation during the pandemic and weight status (BMI, Body Mass Index) were evaluated. Skin colour classification followed the same criteria used by the Brazilian Institute of Geography and Statistics [26] The cut-off points for BMI classification (obesity, overweight, eutrophic/normal weight and underweight) were based on the World Health Organization (WHO) guidelines [27]. Changes in consumption of selected hyperpalatable foods: cakes, sweet cookies, savoury biscuits, packet snacks, ready-to-eat foods (instant noodles, frozen packaged meals, nuggets…), sweets, and sweetened soft drinks, and changes in the three eating habits, during the pandemic were also evaluated.

## Data analysis

Relative and absolute frequencies of the main variables were estimated. The chi-squared test was used to access whether the difference in anxiety symptoms in males and females was statistically significant. To evaluate the consumption of hyperpalatable foods, changes in eating habits and the presence of self-reported symptoms of anxiety and depression, an exploratory structural equation model (SEM) was proposed, considering the sampling weights by course and sex. Firstly, we adjusted the measurement models, which established the latent variables. Confirmatory factor analysis was conducted using latent variables that showed significant factor loadings ($p < 0.05$), with standardized loadings of ≥ 0.40 for food consumption variables [28] and ≥ 0.50 for other variables [29]. There was no missing data.

In this model, the "consumption of hyperpalatable foods" (CONS) was hypothesized as a latent variable composed of seven indicators (observed variables) namely "savoury biscuits", "sweet cookies", "instant meals", "snacks", "sweets" and "sweetened soft drinks". The latent variable "exchange of meals for snacks" (REFL) was constructed based on the answers to the questions regarding meal patterns, namely "replacement of lunch with snacks", "replacement of dinner

with snacks" and "consumption of ready-to-eat meals via delivery". Finally, the latent variable "symptoms of anxiety and depression" (TRAN) was constructed based on the scores of the GAD-7 and PHQ-9.

Subsequently, a multivariate model was adjusted using SEM. Estimates were obtained using the Weighted Least Squares Means and Variance method adjusted by the mean and variance (WLSMV), which is considered the most appropriate due to its robustness against potential violations of the normality assumption in parameter distributions.

To assess the goodness of fit of the models, the following criteria were used: Root Mean Square Error of Approximation (RMSEA) < 0.06, Standardized Root Mean Square Residual (SRMSR) < 0.05, and Comparative Fit Index (CFI), Tucker-Lewis Index (TLI) and Goodness of Fit index (GFI) all > 0.95.

All data processing and statistical analysis was performed in R (version 1.4.1106). The structural equations model was performed using the Lavaan package (version 0.6–11).

### Ethical aspects

The present study was approved by the Research Ethics Committee of the Institute of Social Medicine at the State University of Rio de Janeiro (CAAE 35340820.1.0000.5260). The procedures for carrying out the research were informed to all participants via e-mail and their participation in the study was voluntary. An informed written consent term was obtained from each participant via a specific field on the Google Form questionnaire. A copy of the consent term was also sent to each participant via e-mail. All participants were 18 years of age or older.

### Results

The sample consisted mostly of female individuals (65.2%), and 58.4% were between 18 and 21 years old (mean age of 24.5 years). Most reported social isolation for 14 weeks or more (79.5%) and the vast majority had not been diagnosed with COVID-19 at the time of answering the questionnaire (91.4%) (Table 1).

More than half of the students self-reported clinically significant symptoms of anxiety (53.8%) and symptoms of depression (62.5%). The presence of both comorbidities was also high, with 47.6% of students having self-reported symptoms of both disorders, with no major differences by sex (Table 2).

Regarding the consumption of hyperpalatable foods, the greatest changes were for increased consumption of cakes (44.2%), sweets (44.2%) and sweetened soft drinks (34.6%). More than half of the participants reported replacing dinner with snacks (54.9%) and 46.2% reported an increase in the consumption of ready-to-eat meals via delivery (Table 3).

Nearly all individuals who reported an increase in the frequency of consumption of hyperpalatable foods, and "replace meals with snacks" and "increase the consumption of ready-to-eat meals via delivery" were also part of the group of individuals with clinically significant symptoms of depression or anxiety. The only exceptions were the decrease in consumption of cakes and ready-to-eat meals via delivery among individuals with significant symptoms of anxiety and depression compared with individuals without significant symptoms (Table 4).

In the confirmatory factor analysis, due to the high prevalence of individuals suffering from self-reported symptoms of both disorders, the latent variable *symptoms of anxiety and depression* (TRAN) was used, composed of the scores of the GAD-7 and PHQ-9.

The change in food consumption included the seven hyperpalatable foods (latent variable CONS), and meal and delivery habits (REFL). These two latent variables are better suited to measure changes in students' eating habits than the individual variables that compose them, as the confirmatory factor analysis used in creating a latent variable removes the variance of observed variables unrelated to the construct.

Fig 1 shows the results of the confirmatory factor analysis that operationalized the latent variables "symptoms of anxiety and depression" (TRAN), changes in the "consumption of hyperpalatable foods" (CONS) and "replacement of meals with snacks" (REFL). Modification indices were used to identify potential paths not initially specified, which, if incorporated into the model, could improve its fit indices. From these modification indexes, suggestions were accepted to

**Table 1. Sample size and weighted frequencies* of student characteristics by sex.**

| | | Sex | |
|---|---|---|---|
| | N (%) | Female (%) | Male (%) |
| **Sex** | 771 (100) | 503 (65.2) | 268 (34.8) |
| **Age** | | | |
| 18 - 21 | 450 (58.4) | 274 (54.5) | 176 (65.7) |
| 22 - 25 | 129 (16.7) | 87 (17.3) | 42 (15.7) |
| 26 - 29 | 56 (7.3) | 37 (7.4) | 19 (7.1) |
| ≥ 30 | 136 (17.6) | 105 (20.9) | 31 (11.6) |
| **Marital Status** | | | |
| Single | 649 (84.2) | 415 (82.5) | 234 (87.3) |
| Married | 108 (14.0) | 76 (15.1) | 32 (11.9) |
| Divorced | 12 (1.6) | 10 (2.0) | 2 (0.7) |
| Widowed | 2 (0.3) | 2 (0.4) | 0 (0.0) |
| **Skin colour** | | | |
| White | 469 (60.8) | 295 (58.6) | 174 (64.9) |
| Black | 114 (14.8) | 76 (15.1) | 38 (14.2) |
| Brown | 177 (23.0) | 123 (24.1) | 54 (20.1) |
| Yellow | 11 (1.4) | 9 (1.8) | 2 (0.7) |
| **Isolation Time** | | | |
| ≤ 13 weeks | 141 (18.3) | 92 (18.3) | 49 (18.3) |
| ≥ 14 weeks | 613 (79.5) | 400 (79.5) | 213 (79.5) |
| No social isolation | 17 (2.2) | 11 (2.2) | 6 (2.2) |
| **Weight Status (BMI)[a]** | | | |
| Underweight | 35 (4.6) | 26 (5.2) | 9 (3.4) |
| Eutrophic | 406 (52.9) | 259 (51.5) | 147 (54.9) |
| Overweight | 202 (26.3) | 133 (26.6) | 69 (25.7) |
| Obesity | 125 (16.3) | 82 (16.4) | 43 (16.0) |
| **COVID-19 Diagnosis** | | | |
| Yes | 66 (8.6) | 36 (7.2) | 30 (11.2) |
| No | 705 (91.4) | 467 (92.8) | 238 (88.8) |
| **Income During Pandemic** | | | |
| Increase | 27 (3.5) | 14 (2.8) | 13 (4.9) |
| No change | 312 (40.5) | 193 (38.4) | 119 (44.4) |
| Decrease | 432 (56.0) | 296 (58.8) | 136 (50.7) |

*Weighted by course and sex.

[a]three (3) individuals failed to inform their weight or height and were excluded from the nutritional status calculation.

[b]six (6) individuals failed to report their health status during the pandemic and were excluded from the health status calculation.

insert correlations between the errors of the indexes "savoury biscuits consumption" with "snacks consumption", "savoury biscuits consumption" with "sweet cookies consumption" and "sweets consumption" with "cakes consumption". All the observed variables that composed each item had adequate factorial weights, being ≥ 0.5 for the variable symptoms of anxiety and depression" (TRAN) and ≥ 0.4 for the others. All the factorial weights were significant at $p < 0.05$. Factor loads for the latent variable TRANS were similar in GAD-7 (0.876) and PHQ-9 (0.871). The highest factor load for the latent variable "consumption of hyperpalatable foods" was associated with the consumption of sweet cookies (0.766). The indices of the models presented values considered acceptable, with RMSEA of 0.043, the lower and upper limits of their confidence

**Table 2. Weighted prevalence of symptoms of anxiety and depression.**

| Variables | | Sex | | |
|---|---|---|---|---|
| | N (%) | Female (%) | Male (%) | P-value |
| **Anxiety symptoms[a]** | | | | |
| clinically significant | 415 (53.8) | 272 (54.1) | 143 (53.4) | 0.90 |
| not significant | 356 (46.2) | 231 (45.9) | 125 (46.6) | |
| **Depression symptoms[b]** | | | | |
| clinically significant | 482 (62.5) | 322 (64.0) | 160 (59.7) | 0.27 |
| not significant | 289 (37.5) | 181 (36.0) | 108 (40.3) | |
| **Symptoms of both** | | | | |
| has symptoms of both | 367 (47.6) | 244 (48.5) | 123 (45.9) | 0.54 |
| anxiety without depression | 48 (6.2) | 28 (5.6) | 20 (7.5) | 0.38 |
| depression without anxiety | 115 (14.9) | 78 (15.5) | 37 (13.8) | 0.60 |
| both not significant | 241 (31.3) | 153 (30.4) | 88 (32.8) | 0.54 |

[a]defined as a score of 10 or more on the Generalized Anxiety Disorder 7-item (GAD-7).

[b]defined as a score of 10 or more on the Patient Health Questionnaire 9-item (PHQ-9).

**Table 3. Variation in the consumption of hyperpalatable foods and the pattern of meals.**

| | Frequency | | |
|---|---|---|---|
| | Decrease (%) | No change (%) | Increase (%) |
| **Foods** | | | |
| Savoury biscuits | 216 (28.0) | 384 (49.8) | 171 (22.2) |
| Sweet cookies | 215 (27.9) | 302 (39.2) | 254 (32.9) |
| Cakes | 122 (15.8) | 308 (40.0) | 341 (44.2) |
| Ready-to-eat meals | 200 (25.9) | 357 (46.3) | 214 (27.8) |
| Packaged snacks | 243 (31.5) | 373 (48.4) | 155 (20.1) |
| Sweets | 135 (17.5) | 224 (29.1) | 412 (53.4) |
| Sweetened soft drinks | 200 (25.9) | 304 (39.4) | 267 (34.6) |
| **Meals** | | | |
| Swap lunch for snack | 197 (25.6) | 430 (55.8) | 144 (18.7) |
| Swap dinner for snack | 124 (16.1) | 224 (29.1) | 423 (54.9) |
| Meals via delivery | 139 (18.0) | 276 (35.8) | 356 (46.2) |

intervals were 0.033 and 0.053, and the value of p = 0.87. The SRMR was 0.049, the CFI was 0.991, the TLI was 0.988, and the GFI was 0.992.

The correlation estimate between the latent variables CONS and TRAN was 0.212, the correlation between CONS and REFL was 0.659 and the correlation between REFL and TRAN was 0.223, all of which were statistically significant (p < 0.01) (Fig 1).

The variable "replacement of meals with snacks" (REFL) showed a significant standardized coefficient with the change of "consumption of hyperpalatable foods" (CONS), indicating a change of 0.643 standard deviations in CONS for each one-unit change in REFL. In turn, for each change of one standard deviation in "symptoms of anxiety and depression" (TRAN) we have a direct change of 0.223 standard deviations in REFL. Although TRAN did not show a statistically significant direct standardized coefficient with CONS (P = 0.07), its effect on CONS can be calculated by indirect effect via the magnitude of the REFL value that comes from TRAN. After taking indirect effects into account, the total effect of TRAN

**Table 4. Variation in the consumption of hyperpalatable foods and in the meal pattern according to the presence of symptoms of anxiety and depression.**

| | Presence: | Frequency | | Chi² | Presence: | Frequency | | Chi² |
|---|---|---|---|---|---|---|---|---|
| | Anxiety[a] | Decrease (%) | Increase (%) | P-value | Depression[b] | Decrease (%) | Increase (%) | P-value |
| Savoury biscuits | No | 103 (28.9) | 52 (14.6) | <0.01 | No | 81 (28.0) | 54 (18.7) | 0.2 |
| | Yes | 113 (27.2) | 119 (28.7) | | Yes | 135 (28.0) | 117 (24.3) | |
| Sweet cookies | No | 106 (29.8) | 93 (26.1) | <0.01 | No | 84 (29.1) | 77 (26.6) | <0.01 |
| | Yes | 109 (26.3) | 161 (38.8) | | Yes | 131 (27.2) | 177 (36.7) | |
| Cakes | No | 55 (15.4) | 138 (38.8) | <0.01 | No | 42 (14.5) | 112 (38.8) | <0.01 |
| | Yes | 67 (16.1) | 203 (48.9) | | Yes | 80 (16.6) | 229 (47.5) | |
| Ready-to-eat meals | No | 99 (27.8) | 75 (21.1) | <0.01 | No | 76 (26.3) | 64 (22.1) | 0.02 |
| | Yes | 101 (24.3) | 139 (33.5) | | Yes | 124 (25.7) | 150 (31.1) | |
| Packaged snacks | No | 118 (33.1) | 50 (14.0) | <0.01 | No | 96 (33.2) | 46 (15.9) | 0.08 |
| | Yes | 125 (30.1) | 105 (25.3) | | Yes | 147 (30.5) | 109 (22.6) | |
| Sweets | No | 68 (19.1) | 162 (45.5) | <0.01 | No | 55 (19.0) | 131 (45.3) | <0.01 |
| | Yes | 67 (16.1) | 250 (60.2) | | Yes | 80 (16.6) | 281 (58.3) | |
| Sweetened soft drinks | No | 104 (29.2) | 91 (25.6) | <0.01 | No | 85 (29.4) | 73 (25.3) | <0.01 |
| | Yes | 96 (23.1) | 176 (42.4) | | Yes | 115 (23.9) | 194 (40.2) | |
| Swap lunch for snack | No | 105 (29.5) | 46 (12.9) | <0.01 | No | 83 (28.7) | 32 (11.1) | <0.01 |
| | Yes | 92 (22.2) | 98 (23.6) | | Yes | 114 (23.7) | 112 (23.2) | |
| Swap dinner for snack | No | 60 (16.9) | 165 (46.3) | <0.01 | No | 53 (18.3) | 126 (43.6) | <0.01 |
| | Yes | 64 (15.4) | 258 (62.2) | | Yes | 71 (14.7) | 297 (61.6) | |
| Meals via delivery | No | 64 (18.0) | 151 (42.4) | 0.1 | No | 51 (17.6) | 123 (42.6) | 0.2 |
| | Yes | 75 (18.1) | 205 (49.4) | | Yes | 88 (18.3) | 233 (48.3) | |

on CONS is 0.212 for each change of one standard deviation. Therefore, there is no direct association between the self-reported symptoms of anxiety and/or depression with changes in the consumption of hyperpalatable foods but the effect is mediated by meal replacement with snacks/delivery (Fig 2).

The goodness of fit indices of the models presented values considered acceptable. The RMSEA was 0.043, the SRMR was 0.049, the CFI was 0.991, the TLI was 0.988, and the GFI was 0.992.

## Discussion

This study showed that male and female students with a high prevalence of depression and anxiety symptoms showed increased consumption of hyperpalatable foods, mediated by meal replacement with snacks.

The inability to cope with the responsibilities arising from the transitional phase between adolescence and adulthood can lead to high levels of stress and anxiety in the undergraduate population [30]. Even before the pandemic, this condition was frequently associated with inadequate food intake [31–33], irregular eating patterns – such as skipping meals and replacing them with snacks [30] – and a decline in mental health quality [32,34] During the COVID-19 pandemic, social distancing measures were implemented in Brazil, and the Ministry of Education authorized the replacement of face-to-face classes with digital classes [35]. Such measures have led university students to spend more time at home and reduced their face-to-face social contacts. This behavioural shift has contributed to a further increase in sedentary behaviour [5], reduced physical activity levels, negative impacts on dietary intake [35], and a deterioration in both sleep quality [36,37] and mental health [18,38].Our analysis supports the hypothesis that higher levels of anxiety and depression symptoms increase consumption of hyperpalatable foods, and the structural equation modelling technique allowed to capture the nuances of this relationship, indicating that meal replacement for snacks increased intake of hyperpalatable food.

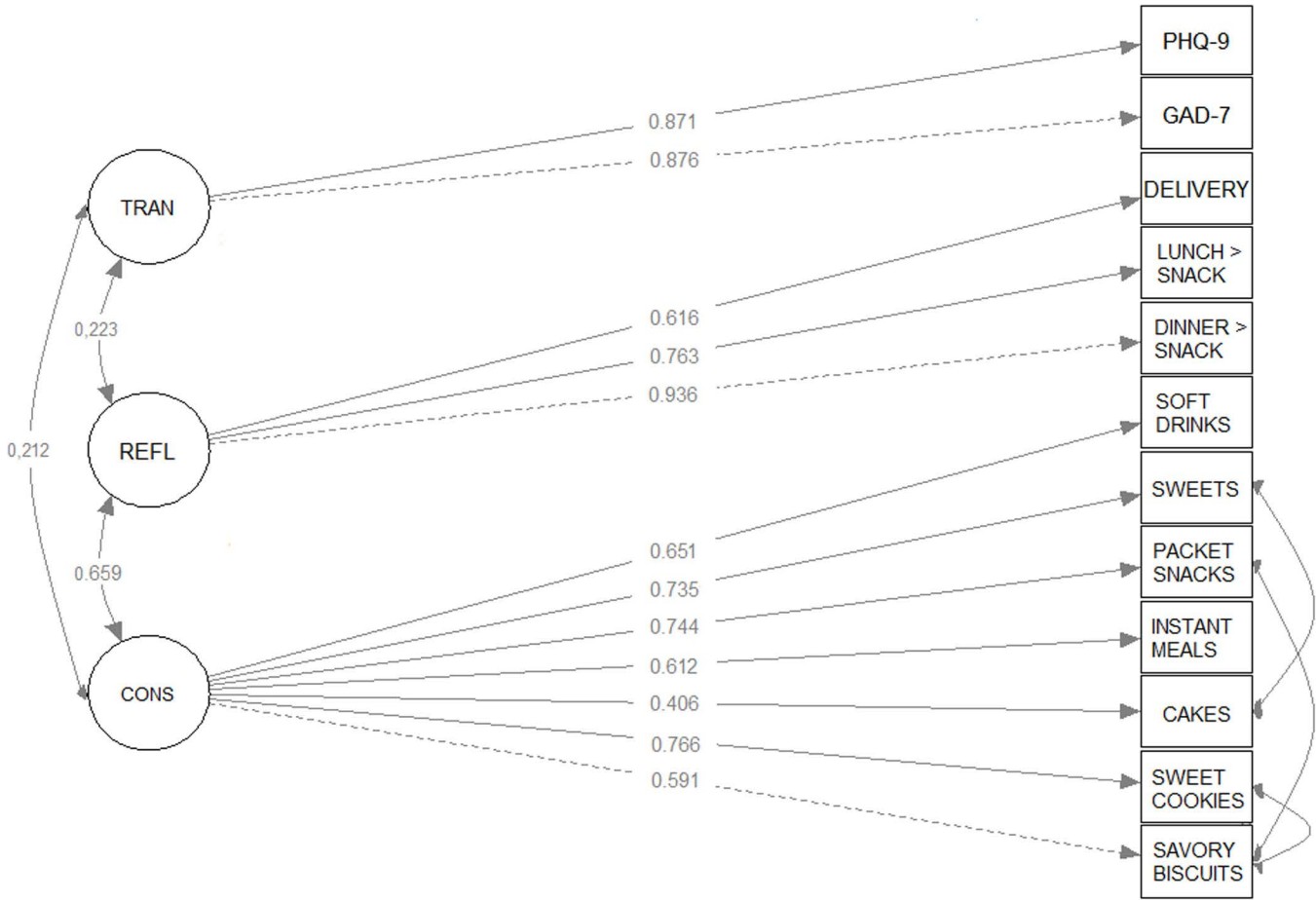

**Fig 1. Diagram of the measurement model adjusted for the constructs: symptoms of anxiety and depression, changes in consumption of hyperpalatable foods and replacement of meals with snacks (n = 771).**

Although the cross-sectional design of the study, longitudinal studies have shown that mental health, particularly anxiety, increases unhealthy food intake [15] and, in this line, our data shows that for every one standard deviation increase in TRAN, there is a 0.212 standard deviation increase in CONS, i.e., higher levels of anxiety and depression symptoms are in turn reflected in greater (positive) change in the intake of the selected hyperpalatable foods. The factor loadings between the latent variables 'meal replacement with snacks' (REFL) and 'consumption of hyperpalatable foods' (CONS) were also high and significant, as expected, given that hyperpalatable foods are predominantly consumed as snacks and often replace traditional meals – which, in Brazil, typically consist of rice, beans, and often meat. [39,40].

The variable that most contributed to the creation of the latent variable REFL, providing the highest factor load (0.936) was "substituting dinner for snacks." This variable showed the highest overall increase among students (Table 3), and remained elevated among those with clinically significant symptoms of anxiety and depression (Table 4). In contrast, the observed variable that contributed with the lowest factor load for REFL creation was "consumption of meals via delivery". The lower factor loading may be explained by the fact that delivery apps offer not only hyperpalatable foods, but also traditional meals and healthier options. However, it is important to note that food delivery apps in Brazil feature menus that are approximately 70% composed of sandwiches, hamburgers, fried snacks and pizzas. These categories are also prominently featured in restaurant advertisements and discounts, often at the expense of traditional meals and vegetables [13].

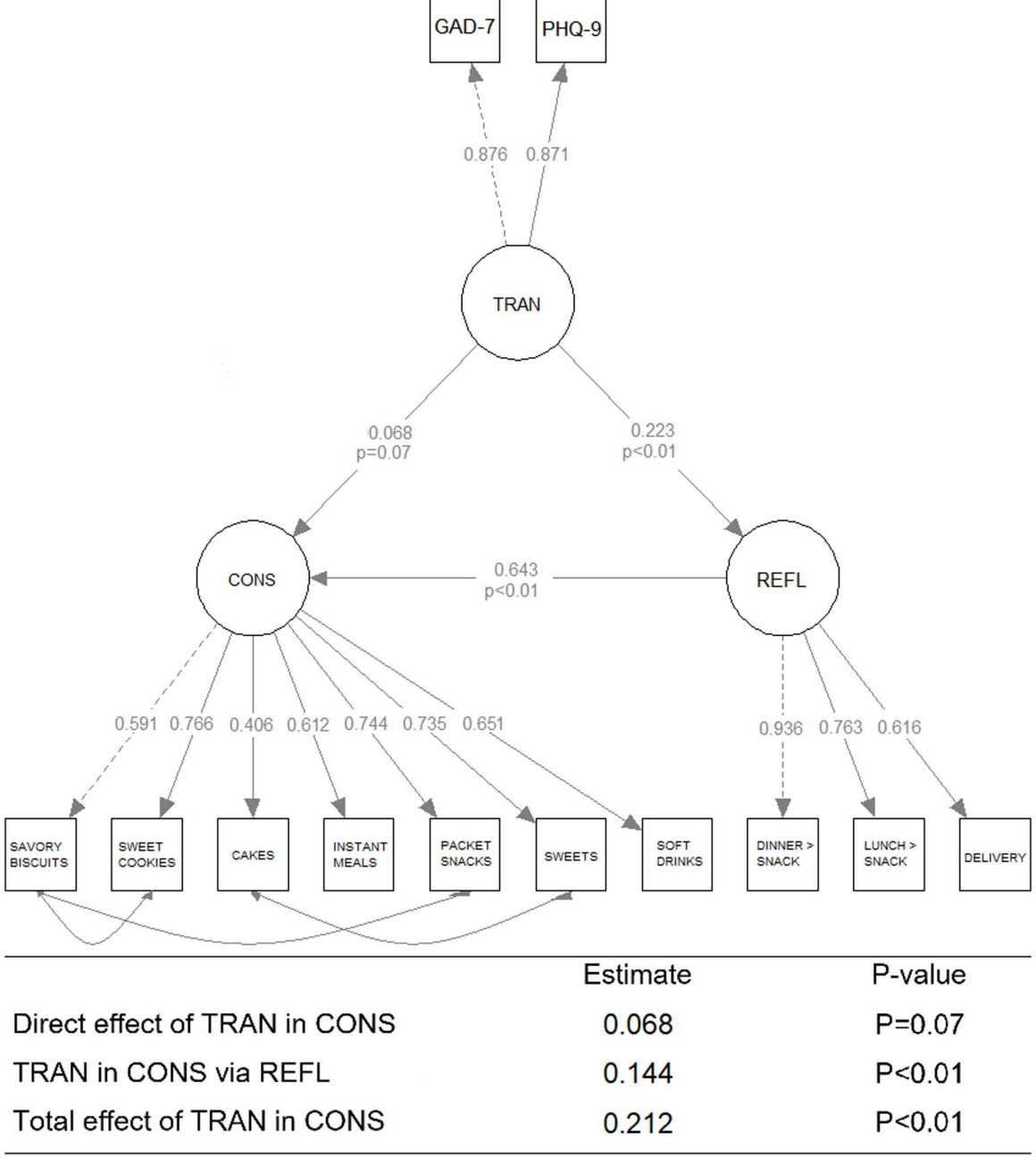

| | Estimate | P-value |
|---|---|---|
| Direct effect of TRAN in CONS | 0.068 | P=0.07 |
| TRAN in CONS via REFL | 0.144 | P<0.01 |
| Total effect of TRAN in CONS | 0.212 | P<0.01 |

**Fig 2. Structural model diagram to assess the influence of the presence of self-reported symptoms of anxiety and/or depression on the change in the consumption habits of hyperpalatable foods and meals. (n=771).**

The majority of the sample (65.2%) was composed of female students, a difference of less than 10% compared to the 55.32% of regularly enrolled female students at the university in 2019 [19]. In contrast to the literature, the authors found no statistically significant differences between the prevalence of anxiety and depression symptoms between female and male students (Table 2). Studies indicate that the lifetime prevalence of depression and anxiety is up to twice as high in

women compared to men [41,42], with the difference tending to diminish, on average, after the age of 65 [43,44]. Specifically among Brazilian college students, previous studies have reported that there is a significant difference between the prevalence of depression and anxiety symptoms between men and women, corroborating the existing literature [16,45,46]. Non-response rate (80.6%) and the online data collection could be a possible explanation, as students experiencing higher levels of depression and/or anxiety may have been less likely to participate in the survey.

The main limitations of this study were: 1) a high non-response rate; 2) the online data collection method, which involved distributing the questionnaire via email, despite the target population typically favouring communication through social media and other collaborative platforms [47]; 3) the study being limited to a single state in Brazil, which may limit the generalizability of the findings to the broader adult population or non-university individuals; However, the university's affirmative action policies, reserving 50% of admissions for Black and low-income students, may help mitigate concerns regarding representativeness. The strengths of this study include: 1) the use of the structural equation modelling, which combines elements of multiple regression, examining multiple dependency relationships, with factorial analysis, allowing the representation of unobserved constructs through multiple indicators [48]. This technique enable us to analyse the multiple levels of interdependence among the three latent variables studied; 2) despite the high non-response rate, respondent distribution across courses was relatively homogeneous, with an average response rate of 19.4% across all courses (see S1 Table), and only a few courses being under- or overrepresented; and 3) access to detailed demographic data – such as the number of students by gender in each course – allowed for the application of sample weights to reduce the impact of non-response bias. Therefore, our findings may be considered representative of the undergraduate student population in Brazilian public universities.

## Conclusion

Our findings underscore a significant association between symptoms of anxiety and depression and the heightened consumption of hyperpalatable foods among university students, largely influenced by the substitution of regular meals with snacks. This relationship suggests that adverse mental health states may lead to detrimental dietary behaviours, carrying potential long-term consequences not only for weight management and metabolic health but also for overall psychological wellbeing. From a practical standpoint, universities and public health agencies should consider implementing screening programmes for mental health symptoms and develop targeted nutritional interventions that address snack-based eating patterns. These strategies may range from psychoeducational workshops on stress management and healthy eating to policy initiatives aimed at regulating the availability of low-nutrition foods on campus.

This study was limited by a high non-response rate and an online data collection approach, factors that may restrict the representativeness of the findings. Future research should consider adopting longitudinal designs to elucidate whether these eating behaviours and mental health issues persist beyond the pandemic context. It would also be worthwhile to explore a broader range of university populations in different geographical and socioeconomic settings, facilitating the generalisation of the results. Strengthening evidence in this area can inform more robust public health policies and institutional practices, ultimately contributing to healthier lifestyles and improved psychological outcomes for students.

## Supporting information

**S1 Table. Frequency of response and percentage of students per course.**
(XLSX)

## Acknowledgments

None.

## Author contributions

**Conceptualization:** Marcus Verly-Miguel, Claudia de Souza Lopes, Jade Veloso Freitas, Rosely Sichieri.

**Data curation:** Marcus Verly-Miguel, Magno Conceição Garcia, Marcio Candeias Marques, Rosely Sichieri.

**Formal analysis:** Marcus Verly-Miguel, Magno Conceição Garcia, Marcio Candeias Marques.

**Funding acquisition:** Claudia de Souza Lopes, Vitor Barreto Paravidino, Rosely Sichieri.

**Investigation:** Vitor Barreto Paravidino, Rosely Sichieri.

**Methodology:** Marcus Verly-Miguel, Claudia de Souza Lopes, Marcio Candeias Marques, Vitor Barreto Paravidino, Rosely Sichieri.

**Project administration:** Marcus Verly-Miguel, Vitor Barreto Paravidino, Rosely Sichieri.

**Resources:** Marcus Verly-Miguel, Magno Conceição Garcia, Vitor Barreto Paravidino.

**Software:** Marcus Verly-Miguel, Magno Conceição Garcia, Marcio Candeias Marques.

**Supervision:** Claudia de Souza Lopes, Rosely Sichieri.

**Validation:** Marcus Verly-Miguel, Claudia de Souza Lopes, Marcio Candeias Marques, Rosely Sichieri.

**Visualization:** Marcus Verly-Miguel, Magno Conceição Garcia, Marcio Candeias Marques.

**Writing – original draft:** Marcus Verly-Miguel, Magno Conceição Garcia, Jade Veloso Freitas, Vitor Barreto Paravidino.

**Writing – review & editing:** Marcus Verly-Miguel, Claudia de Souza Lopes, Rosely Sichieri.

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
