## [Editor Report · Decision Letter 0]

Dear Dr. Verly-Miguel,

Thank you for submitting your manuscript to PLOS ONE. After careful consideration, we feel that it has merit but does not fully meet PLOS ONE’s publication criteria as it currently stands. Therefore, we invite you to submit a revised version of the manuscript that addresses the points raised during the review process.

We look forward to receiving your revised manuscript.

Kind regards,

Bruno Luciano Carneiro Alves de Oliveira

Academic Editor

PLOS ONE

Journal Requirements:

"Funding received according to the grant n° 255937: Emergencial Action COVID-19 - Call C - Support for Research Network Projects on SARS-CoV-2/COVID-19. Funding provided by Carlos Chagas Filho Foundation (FAPERJ)."

4. Please upload a copy of Figure 4, to which you refer in your text on page 16. If the figure is no longer to be included as part of the submission please remove all reference to it within the text.

**Additional Editor Comments:**

Introduction

As the results are presented by gender, I suggest considering this difference in the introduction.

Methods

In what period, month, year and wave of COVID-19? Was this research done? Did we already have a vaccine? If so, may or may not this have affected the perception of the pandemic and thus anxiety and depression or eating habits?

No sample calculation was carried out?

What is the sampling power to capture differences in relation to population size?

The authors must consider that they are two instruments for the epidemiological screening of anxiety and depression and not for the clinical diagnosis of these conditions.

The sociodemographic variables that make up the article, especially in table 1, were not described in the methods section. For example, area of study and what corresponds.

Was the research approved by a research ethics committee in Brazil? What number indicates this approval?

Discussion

Can it be said that mental health problems were identified with the questionnaires administered? Or would it be better to talk about the symptoms of the conditions being researched?

There are other limitations that the authors did not mention: Information, memory, confirmation bias. This is tracking data. Sample size. University students from a single state in Brazil, with a financial standard that may define a dietary pattern that may be different from the general and non-university adult population.

---

## [Author Response · Author response to Decision Letter 1]

4 May 2024

Journal Requirements

A: Style requirements revised.

"Funding received according to the grant n° 255937: Emergencial Action COVID-19 - Call C - Support for Research Network Projects on SARS-CoV-2/COVID-19. Funding provided by Carlos Chagas Filho Foundation (FAPERJ)."

Please state what role the funders took in the study. If the funders had no role, please state: ""The funders had no role in study design, data collection and analysis, decision to publish, or preparation of the manuscript."" If this statement is not correct you must amend it as needed. Please include this amended Role of Funder statement in your cover letter; we will change the online submission form on your behalf.

A: Amended Role of Funded statement added to the Cover Letter.

A: Full name of the IRB added; the ethics statement contains the following information regarding the written consent “An informed written consent term was obtained from each participant via a specific field on the Google Form questionnaire” that can be found on lines 158 and 159 of the Revised Manuscript with Track Changes.

4. Please upload a copy of Figure 4, to which you refer in your text on page 16. If the figure is no longer to be included as part of the submission please remove all reference to it within the text.

A: We meant to write “Table 4” and wrote “Figure 4” by accident. Corrected.

Additional Editor Comments

Introduction

1. As the results are presented by gender, I suggest considering this difference in the introduction.

A: We added a new mention and reference regarding gender that can be seen on lines 58 and 59 of the Revised Manuscript with Track Changes.

Methods

2. In what period, month, year and wave of COVID-19 was this research done? Did we already have a vaccine? If so, may or may not this have affected the perception of the pandemic and thus anxiety and depression or eating habits?

A: Our data was collected between August 6, 2020 and March 13, 2021, as stated on line 76 of the Revised Manuscript with Track Changes, a period that comprehends the second wave of COVID-19. No vaccines were available in Brazil. New text was added stating this information (lines 76 and 77).

3. No sample calculation was carried out? What is the sampling power to capture differences in relation to population size?

A: The sampling power to capture differences in relation to population size was 0.8, the sample size using a proportion of 50% and an error of 0.05 was 384. This information was added on lines 74 and 75 of the Revised Manuscript with Track Changes.

4. The authors must consider that they are two instruments for the epidemiological screening of anxiety and depression and not for the clinical diagnosis of these conditions.

A: We made sure to point out in the methodology section of the text that our instruments are used to investigate the presence of self-reported anxiety symptoms. No mentions of clinical diagnosis of anxiety and/or depression were made throughout the text.

5. The sociodemographic variables that make up the article, especially in table 1, were not described in the methods section. For example, area of study and what corresponds.

A: After careful consideration the authors have decided to remove the “knowledge area” variable from Table 1, since this variable wasn’t used on our analysis or anywhere else in the text. Regarding the other sociodemographic variables, we identified that “race” and “weight status (BMI)” may elicit some doubt in the reader and further explanation was provided on lines 117 to 120 of the Revised Manuscript with Track Changes.

Now, weight status was defined as BMI classification of the WHO and race/ skin color was defined in agreement with the Brazilian Bureau of Census.

6. Was the research approved by a research ethics committee in Brazil? What number indicates this approval?

A: Yes, our study was approved by the Research Ethics Committee of the Institute of Social Medicine at the State University of Rio de Janeiro, and it received approval based on opinion number 4,189,209. This information was added on lines 154 to 156 of the Revised Manuscript with Track Changes.

Discussion

7. Can it be said that mental health problems were identified with the questionnaires administered? Or would it be better to talk about the symptoms of the conditions being researched?

A: Mental health problems can not be identified with the questionnaire administered, only symptoms. We have removed this statement on lines 247 and 248 of the Revised Manuscript with Track Changes.

8. There are other limitations that the authors did not mention: Information, memory, confirmation bias. This is tracking data. Sample size. University students from a single state in Brazil, with a financial standard that may define a dietary pattern that may be different from the general and non-university adult population.

A: Since the study was conducted during the first waves of the pandemic, the auctors don’t believe that memory and confirmation bias are an issue. The reviewer made a good point on the possible lack of representability of our sample and we added a third limitation pointing this out, located between lines 302 and 306 of the Revised Manuscript with Track Changes. Also, we added the argument used by the editor.

---

## [Decision Letter · Decision Letter 1]

Dear Dr. Verly-Miguel,

Thank you for submitting your manuscript to PLOS ONE. After careful consideration, we feel that it has merit but does not fully meet PLOS ONE’s publication criteria as it currently stands. Therefore, we invite you to submit a revised version of the manuscript that addresses the points raised during the review process.

Comments follow at the end of the email.

We look forward to receiving your revised manuscript.

Kind regards,

Sandro Vieira Soares, Ph.D.

Academic Editor

PLOS ONE

Journal Requirements:

Additional Editor Comments:

Based on the reviewers' reports, I have carefully evaluated the manuscript and would like to recommend a minor revision. While the manuscript is of high quality and contributes significantly to the field, there are a few points raised by the reviewers that require clarification or adjustment before proceeding with publication.

Reviewer 2:

The conclusion of the article is concise but does not sufficiently address the main findings and implications of the study. For a manuscript submitted to a high-impact journal like PLOS ONE, it is recommended that the Conclusion section be more comprehensive, reflecting the following points:

Summary of key findings: The conclusion should synthesize the main findings discussed in the paper, particularly the significant relationship between symptoms of anxiety and depression and the increased consumption of hyperpalatable foods, mediated by meal substitution with snacks. This is crucial to reinforce the importance of the discoveries.

Practical and scientific implications: The impact of the results on the mental and behavioral health of university students should be emphasized more strongly. For example, the conclusion could discuss the potential long-term implications of these eating habits for both the mental and physical health of the university population.

Study limitations: While the limitations have been addressed in the discussion, it would be important to briefly mention them in the conclusion, particularly the high non-response rate and the online data collection method, which may affect the generalizability of the results.

Suggestions for future research: Including a brief mention of future research directions could enhance the conclusion. For instance, suggesting longitudinal studies to investigate the continuity of these eating behaviors post-pandemic or applying the study to other university populations from different geographical and socioeconomic contexts.

A more robust conclusion would help better contextualize the results and provide a clearer understanding of the study’s relevance to the field of public health and eating psychology. Additionally, it could strengthen the paper’s impact on its readership.

Reviewer 3:

In the process of composing and describing variables, the authors treat the self-reported manifestations of anxiety and depression symptoms as mental disorders. The reviewer considers that this expression can lead to confusion to the less experienced reader on the subject, since it would characterize the possibility of considering the presence of mental disorders by the set of answers to the questionnaires. Although it is known by the authors that this is not the main objective of their work, the reviewer is careful to characterize the possible confusion on the part of the reader. It is therefore suggested that the authors may replace the expression mental disorders with self-reported symptoms of anxiety and/or depression, or even with responses of self-reported symptoms of anxiety and/or depression.

The discussion item presents a close relationship between the objectives of the investigation and the findings in the international and national literature regarding the possibility of explaining the characteristics of food choice and manifestations, whether demographic, school, gender, or even from the interference of symptomatic manifestations of anxiety and depression. The conclusion can be briefly expanded to account for the main aspects, including limitations, of the study, even if they are presented in the previous topic.

The study deals with a sensitive, important, current and relevant theme in the context of human development, collective health, nutritional health as well as social processes related to the university environment, in a specific period that was the COVID-19 pandemic. For these reasons, the reviewer reinforces the need for and importance of the manuscript being clear and not generating doubts, while encouraging authors to rectify it.

Reviewer 4:

Title – I suggest including a mention of covid-19.

Abstract – The sentence about the analysis method is after the results. I suggest inverting it.

Introduction – I suggest inserting the percentage value next to the numbers 3.973 and 771 (line 72)

Methods

• Sample weighting – There is no need to repeat the values 5.9 and 18.7 (lines 83 and 84)

• Questionnaire: If possible, insert a bibliographic reference in line 88, when it is mentioned that 29 food-items frequently cited in dietary surveys in Brazil were included or inform the types of foods included.

• Study variables – check if the translation of the meaning of the acronym BMI (weight status?) is correct

• Data Analysis – Press enter in the sentence that begins with ‘To assess the goodness...’ (line 145)

Results

Standardize table formatting (see detailed item below).

Complete the value after the decimal point in the text. It was not presented in line 182 (46%)

There are sentences that are in this section, but that should be taken to the methods section. Lines: 204 to 210 ‘Modification indexes were used... with cakes consumption’. Or for the Discussion section – lines 196 to 201

Discussion

There is no need to repeat the results presented previously. If they are maintained, leave only one space after the comma. Example: lines 286 – (65.24%), 287, 291, 292, 310

Conclusion – Insert recommendations for health strategies or policies that can change this reality.

References – Check if the way of presenting the references is in accordance with the journal’s instructions. There are some that contain ellipses in the presentation of the authors/co-authors. Examples: 15, 18, 20, 33 and 35.

Tables

The titles of all tables should be complemented with the location and year.

(,) and (.) are being used to present the percentage values. Correcting

There are percentage values where a decimal point was not added (example Married – table 1)

The tables need to be reformatted as tables. There are several corrections to be made. Example: dividing lines should not be used below the variables that correspond to the columns.

It is not necessary to repeat (p=) in all the values that appear in the p-value column

Reviewers' comments:

Reviewer's Responses to Questions

**Comments to the Author**

Reviewer #1: All comments have been addressed

Reviewer #2: All comments have been addressed

Reviewer #3: (No Response)

Reviewer #4: All comments have been addressed

2. Is the manuscript technically sound, and do the data support the conclusions?

Reviewer #1: Yes

Reviewer #2: Yes

Reviewer #3: Yes

Reviewer #4: Yes

3. Has the statistical analysis been performed appropriately and rigorously?

Reviewer #1: Yes

Reviewer #2: Yes

Reviewer #3: Yes

Reviewer #4: N/A

4. Have the authors made all data underlying the findings in their manuscript fully available?

Reviewer #1: No

Reviewer #2: (No Response)

Reviewer #3: Yes

Reviewer #4: Yes

5. Is the manuscript presented in an intelligible fashion and written in standard English?

Reviewer #1: Yes

Reviewer #2: (No Response)

Reviewer #3: Yes

Reviewer #4: Yes

Reviewer #1: I would like to commend the authors for the excellent work presented. The study is well-executed, relevant, and addresses a timely and significant issue in the context of mental health and eating habits during the COVID-19 pandemic. The methodology employed was appropriate and robust, and the data analysis, particularly the use of structural equation modeling, effectively explored the interrelationships between symptoms of anxiety and depression and the consumption of hyperpalatable foods. The results are relevant and make a valuable contribution to understanding the impact of the pandemic on the eating behaviors of Brazilian university students, providing important insights both for academia and public health policies.

The conclusion of the article is concise but does not sufficiently address the main findings and implications of the study. For a manuscript submitted to a high-impact journal like PLOS ONE, it is recommended that the Conclusion section be more comprehensive, reflecting the following points:

Summary of key findings: The conclusion should synthesize the main findings discussed in the paper, particularly the significant relationship between symptoms of anxiety and depression and the increased consumption of hyperpalatable foods, mediated by meal substitution with snacks. This is crucial to reinforce the importance of the discoveries.

Practical and scientific implications: The impact of the results on the mental and behavioral health of university students should be emphasized more strongly. For example, the conclusion could discuss the potential long-term implications of these eating habits for both the mental and physical health of the university population.

Study limitations: While the limitations have been addressed in the discussion, it would be important to briefly mention them in the conclusion, particularly the high non-response rate and the online data collection method, which may affect the generalizability of the results.

Suggestions for future research: Including a brief mention of future research directions could enhance the conclusion. For instance, suggesting longitudinal studies to investigate the continuity of these eating behaviors post-pandemic or applying the study to other university populations from different geographical and socioeconomic contexts.

A more robust conclusion would help better contextualize the results and provide a clearer understanding of the study’s relevance to the field of public health and eating psychology. Additionally, it could strengthen the paper’s impact on its readership.

Reviewer #2: PONE-D-24-00167_R1 – Review - Anxiety and depression negatively affected the eating habits of university students in Brazil

In this review, I was only responsible for assessing whether the authors had responded to the suggestions for changes made by previous reviewers.

I found that all suggestions were accepted and questions answered. I therefore consider the manuscript accepted for publication.

I would like to take this opportunity to also make some suggestions, only for some additional improvements. There is no need to return it to me.

Title – I suggest including a mention of covid-19.

Abstract – The sentence about the analysis method is after the results. I suggest inverting it.

Introduction – I suggest inserting the percentage value next to the numbers 3.973 and 771 (line 72)

Methods

• Sample weighting – There is no need to repeat the values 5.9 and 18.7 (lines 83 and 84)

• Questionnaire: If possible, insert a bibliographic reference in line 88, when it is mentioned that 29 food-items frequently cited in dietary surveys in Brazil were included or inform the types of foods included.

• Study variables – check if the translation of the meaning of the acronym BMI (weight status?) is correct

• Data Analysis – Press enter in the sentence that begins with ‘To assess the goodness...’ (line 145)

Results

Standardize table formatting (see detailed item below).

Complete the value after the decimal point in the text. It was not presented in line 182 (46%)

There are sentences that are in this section, but that should be taken to the methods section. Lines: 204 to 210 ‘Modification indexes were used... with cakes consumption’. Or for the Discussion section – lines 196 to 201

Discussion

There is no need to repeat the results presented previously. If they are maintained, leave only one space after the comma. Example: lines 286 – (65.24%), 287, 291, 292, 310

Conclusion – Insert recommendations for health strategies or policies that can change this reality.

References – Check if the way of presenting the references is in accordance with the journal’s instructions. There are some that contain ellipses in the presentation of the authors/co-authors. Examples: 15, 18, 20, 33 and 35.

Tables

The titles of all tables should be complemented with the location and year.

(,) and (.) are being used to present the percentage values. Correcting

There are percentage values where a decimal point was not added (example Married – table 1)

The tables need to be reformatted as tables. There are several corrections to be made. Example: dividing lines should not be used below the variables that correspond to the columns.

It is not necessary to repeat (p=) in all the values that appear in the p-value column



Reviewer #3: The manuscript presented refers to a study to monitor eating habits in a sample of university students from a state in the southeastern region of Brazil, during the COVID-19 pandemic period. The authors use a set of varied, pertinent and compatible references to the theme and current events. The manuscript, written in a clear and objective way, seeks to establish possible associations between the process of food choice and symptomatic manifestations of anxiety and depression. The authors present a methodological set capable of meeting this objective, detailing the aspects of sampling, instrumental, statistical analysis and procedures, thus characterizing the possibility of reproduction of the present study. The statistical processes described are compatible with those recommended by international literature and meet the objective and characteristics of the defined variables. The use of tables and charts is pertinent and compatible with the need of illustrating the information to the reader of the findings.

In the process of composing and describing variables, the authors treat the self-reported manifestations of anxiety and depression symptoms as mental disorders. The reviewer considers that this expression can lead to confusion to the less experienced reader on the subject, since it would characterize the possibility of considering the presence of mental disorders by the set of answers to the questionnaires. Although it is known by the authors that this is not the main objective of their work, the reviewer is careful to characterize the possible confusion on the part of the reader. It is therefore suggested that the authors may replace the expression mental disorders with self-reported symptoms of anxiety and/or depression, or even with responses of self-reported symptoms of anxiety and/or depression.

The discussion item presents a close relationship between the objectives of the investigation and the findings in the international and national literature regarding the possibility of explaining the characteristics of food choice and manifestations, whether demographic, school, gender, or even from the interference of symptomatic manifestations of anxiety and depression. The conclusion can be briefly expanded to account for the main aspects, including limitations, of the study, even if they are presented in the previous topic.

The study deals with a sensitive, important, current and relevant theme in the context of human development, collective health, nutritional health as well as social processes related to the university environment, in a specific period that was the COVID-19 pandemic. For these reasons, the reviewer reinforces the need for and importance of the manuscript being clear and not generating doubts, while encouraging authors to rectify it.

Reviewer #4: Manuscript Review Report – PLOS ONE

Dear Editors,

After a detailed analysis of the submitted manuscript, I highlight the following observations regarding adherence to PLOS ONE's criteria and the corrections made by the authors:

1. Manuscript Style:

It was requested that the manuscript comply with PLOS ONE's style requirements. The authors reported that they revised the text according to the required guidelines, ensuring greater clarity and standardization.

2. Funding Statement:

The journal requested that the authors specify the role of funders in the study. The following statement was included: "The funders had no role in the study design, data collection and analysis, decision to publish, or preparation of the manuscript." This information is now clear and in accordance with the guidelines.

3. Ethics Statement:

The authors were asked to include the complete ethics statement in the Methods section, specifying the responsible committee and participant consent. They confirmed that the study was approved by the Research Ethics Committee of the Institute of Social Medicine at the State University of Rio de Janeiro, under opinion number 4.189.209.

4. Figures and Tables Correction:

An error was identified in the mention of a nonexistent figure. The authors clarified that the reference to “Figure 4” was incorrect and that it actually referred to “Table 4.” The manuscript was appropriately corrected.

5. Methodological Adjustments:

Reviewers requested more details about the data collection period and the vaccination situation in Brazil during the study. The authors clarified that the data were collected between August 6, 2020, and March 13, 2021, during the second wave of COVID-19, when vaccines were not yet available in the country.

6. Sample Size Calculation:

It was questioned whether the sample size calculation was performed and what the statistical power was to detect significant differences. The authors explained that the statistical power was 0.8, considering a minimum sample of 384 participants, with a 50% proportion and a 0.05 error.

7. Distinction Between Screening and Diagnosis:

Reviewers emphasized that the instruments used were intended for epidemiological screening, not clinical diagnosis. The authors adjusted the text to clarify that the approach was based on the identification of self-reported symptoms.

8. Sociodemographic Variables:

Reviewers noted that some variables, such as the area of study, were not described in the Methods section. In response, the authors removed this variable from Table 1, as it was not used in subsequent analyses.

9. Study Limitations:

Limitations were pointed out, such as recall bias, confirmation bias, and the representativeness of the sample, composed of university students from a single Brazilian state. The authors acknowledged these limitations and included the observations in the appropriate section of the manuscript.

Conclusion:

The authors addressed the reviewers' requests, making the necessary corrections regarding technical, methodological, and ethical aspects. Given the changes made and the clarity of the justifications provided, I consider the manuscript suitable for publication, provided that the editors are satisfied with the revisions made.

Sincerely,

Andréa Cristina Alves

PLOS ONE Reviewer

**Do you want your identity to be public for this peer review?** For information about this choice, including consent withdrawal, please see our Privacy Policy

Reviewer #1: No

Reviewer #2: No

Reviewer #3: **Yes: ** JOAO CARLOS ALCHIERI

Reviewer #4: No

---

## [Author Response · Author response to Decision Letter 2]

5 May 2025

Depression, anxiety and change in eating habits during the COVID-19 pandemic in Brazilian university students [PONE-D-24-00167R1]

Dear Editor,

We are pleased to resubmit our revised manuscript now entitled “Depression, anxiety and change in eating habits during the COVID-19 pandemic in Brazilian university students” [PONE-D-24-00167R1], for consideration for publication in Plos One.

We would like to express our sincere appreciation to you and the reviewers for your careful reading of our manuscript and for the insightful comments and suggestions that helped improve the quality and clarity of our work.

A detailed point-by-point response to each reviewer follows below, outlining how their feedback was integrated.

We believe the revisions substantially strengthen the manuscript and bring greater clarity to our contribution regarding eating behaviours, psychological symptoms, and public health challenges among university students during the COVID-19 pandemic.

We thank you again for your time and consideration and look forward to your feedback.

Sincerely,

Marcus Vinicius Barbosa Verly-Miguel (on behalf of all co-authors)

Institute of Social Medicine, State University of Rio de Janeiro

Reviewer 2:

The conclusion of the article is concise but does not sufficiently address the main findings and implications of the study. For a manuscript submitted to a high-impact journal like PLOS ONE, it is recommended that the Conclusion section be more comprehensive, reflecting the following points:

Summary of key findings: The conclusion should synthesize the main findings discussed in the paper, particularly the significant relationship between symptoms of anxiety and depression and the increased consumption of hyperpalatable foods, mediated by meal substitution with snacks. This is crucial to reinforce the importance of the discoveries.

Practical and scientific implications: The impact of the results on the mental and behavioral health of university students should be emphasized more strongly. For example, the conclusion could discuss the potential long-term implications of these eating habits for both the mental and physical health of the university population.

Study limitations: While the limitations have been addressed in the discussion, it would be important to briefly mention them in the conclusion, particularly the high non-response rate and the online data collection method, which may affect the generalizability of the results.

Suggestions for future research: Including a brief mention of future research directions could enhance the conclusion. For instance, suggesting longitudinal studies to investigate the continuity of these eating behaviours post-pandemic or applying the study to other university populations from different geographical and socioeconomic contexts.

A more robust conclusion would help better contextualize the results and provide a clearer understanding of the study’s relevance to the field of public health and eating psychology. Additionally, it could strengthen the paper’s impact on its readership.

Response to Reviewer 2:

We appreciate your valuable recommendation. In response, we have expanded the Conclusion section to approximately two paragraphs, addressing the key findings, practical and scientific implications, study limitations, and suggestions for future research (see lines 317–335 in the revised manuscript). Below is a summary of how each of your concerns has been addressed:

• Summary of Key Findings: We synthesised the main results of the study, emphasizing the significant association between symptoms of anxiety/depression and increased intake of hyperpalatable foods, mediated by the replacement of regular meals with snacks.

• Practical and Scientific Implications: We highlighted the potential long-term impacts for both physical and mental health and emphasized the need for university-based programmes and broader public health interventions to mitigate these maladaptive dietary behaviours.

• Study Limitations: We briefly addressed the primary limitations, including the high non-response rate and the use of online data collection, which may limit the generalizability of the findings.

• Suggestions for Future Research: We suggested longitudinal studies to assess the persistence of these behaviours beyond the Covid-19 pandemic and highlighted the importance of expanding research to more diverse geographic regions and student populations.

Reviewer 3:

In the process of composing and describing variables, the authors treat the self-reported manifestations of anxiety and depression symptoms as mental disorders. The reviewer considers that this expression can lead to confusion to the less experienced reader on the subject, since it would characterize the possibility of considering the presence of mental disorders by the set of answers to the questionnaires. Although it is known by the authors that this is not the main objective of their work, the reviewer is careful to characterize the possible confusion on the part of the reader. It is therefore suggested that the authors may replace the expression mental disorders with self-reported symptoms of anxiety and/or depression, or even with responses of self-reported symptoms of anxiety and/or depression.

The discussion item presents a close relationship between the objectives of the investigation and the findings in the international and national literature regarding the possibility of explaining the characteristics of food choice and manifestations, whether demographic, school, gender, or even from the interference of symptomatic manifestations of anxiety and depression. The conclusion can be briefly expanded to account for the main aspects, including limitations, of the study, even if they are presented in the previous topic.

The study deals with a sensitive, important, current and relevant theme in the context of human development, collective health, nutritional health as well as social processes related to the university environment, in a specific period that was the COVID-19 pandemic. For these reasons, the reviewer reinforces the need for and importance of the manuscript being clear and not generating doubts, while encouraging authors to rectify it.

Response to Reviewer 3:

Mental Disorders: Thank you for highlighting this concern. In response, we have revised the manuscript to replace all instances of the term “mental disorders” with either “self-reported symptoms of anxiety and/or depression” or “symptoms of anxiety and depression”, as appropriate. This change was made to prevent any misinterpretation regarding clinical diagnosis and to more accurately reflect the self-reported nature of the data collected. We believe this adjustment enhances the clarity and conceptual precision of the manuscript, particularly for readers less familiar with the topic.

Conclusion Section: We have expanded the Conclusion to more explicitly summarize how the study’s objectives were addressed, while also reiterating key methodological limitations (see lines 317–335). These changes aim to reinforce the study’s contributions and ensure a more comprehensive and transparent final synthesis.

Reviewer 4:

In this review, I was only responsible for assessing whether the authors had responded to the suggestions for changes made by previous reviewers.

I found that all suggestions were accepted and questions answered. I therefore consider the manuscript accepted for publication.

I would like to take this opportunity to also make some suggestions, only for some additional improvements. There is no need to return it to me.

Title – I suggest including a mention of covid-19.

Abstract – The sentence about the analysis method is after the results. I suggest inverting it.

Introduction – I suggest inserting the percentage value next to the numbers 3.973 and 771 (line 72)

Methods

• Sample weighting – There is no need to repeat the values 5.9 and 18.7 (lines 83 and 84)

• Questionnaire: If possible, insert a bibliographic reference in line 88, when it is mentioned that 29 food-items frequently cited in dietary surveys in Brazil were included or inform the types of foods included.

• Study variables – check if the translation of the meaning of the acronym BMI (weight status?) is correct

• Data Analysis – Press enter in the sentence that begins with ‘To assess the goodness...’ (line 145)

Results

• Standardize table formatting (see detailed item below).

• Complete the value after the decimal point in the text. It was not presented in line 182 (46%)

• There are sentences that are in this section, but that should be taken to the methods section. Lines: 204 to 210 ‘Modification indexes were used... with cakes consumption’. Or for the Discussion section – lines 196 to 201

Discussion

• There is no need to repeat the results presented previously. If they are maintained, leave only one space after the comma. Example: lines 286 – (65.24%), 287, 291, 292, 310

Conclusion – Insert recommendations for health strategies or policies that can change this reality.

References – Check if the way of presenting the references is in accordance with the journal’s instructions. There are some that contain ellipses in the presentation of the authors/co-authors. Examples: 15, 18, 20, 33 and 35.

Tables

• The titles of all tables should be complemented with the location and year.

• (,) and (.) are being used to present the percentage values. Correcting

• There are percentage values where a decimal point was not added (example Married – table 1)

• The tables need to be reformatted as tables. There are several corrections to be made. Example: dividing lines should not be used below the variables that correspond to the columns.

• It is not necessary to repeat (p=) in all the values that appear in the p-value column

Response to Reviewer 4:

We thank Reviewer 4 for confirming that all previous suggestions were addressed satisfactorily and for providing additional thoughtful recommendations. Below we outline how each suggestion has been considered and implemented:

• Title: We appreciate the suggestion to include “COVID-19” in the title, which now is “Depression, anxiety and change in eating habits during the COVID-19 pandemic in Brazilian university students”.

• Abstract: The current order reflects the logical flow of structural equation modelling. We presented the results first to describe the observed associations, followed by a description of the modelling approach used to support these findings. This sequence is intended to ensure clarity regarding the data patterns before introducing the analytical framework and underpins them. However, we remain open to revising this order should the editorial board prefer the conventional structure.

• Methods: Population and study design: Percentage values were added as suggested (see line 72)

• Methods: Sample Weighting: The decision to repeat these values was intentionally retained to aid reader understanding of the sample distribution and weighting rationale.

• Questionnaire: Added an extra bibliographic reference, as suggested by the reviewer (see line 90, reference 22)

• Study variables: We revised the description of BMI to ensure accurate terminology and clarity for international readers (see line 118)

• Data Analysis: A line break was added to improve readability, as suggested (see lines 147-148)

Results:

a) All tables were standardized, according to formatting guidelines.

b) Missing decimal point was added (see line 184)

c) Moving sentences: Regarding the reviewer’s suggestion to relocate sentences (lines 196–210), we chose to retain this content in the Results section to maintain analytical transparency. Presenting the use of modification indices and the decisions behind model adjustments alongside the results allows readers to follow the evolution of the analysis and understand how key methodological choices shaped the findings.

Discussion: We maintained select results in the Discussion section as a contextual reminder for readers. However, we implemented the reviewer’s formatting suggestion by ensuring consistent spacing after commas (see lines 286-294 and line 309)

Conclusion: We expanded the Conclusion to include specific and policy-oriented recommendations for universities and public health stakeholders, with an emphasis on promoting healthier food environments (see lines 317–335).

References: Thank you for bringing this matter to our attention. All references have been revised to align with the journal’s guidelines. Specifically, ellipses were replaced with “et al.” after the sixth author’s name.

Tables:

a) Location and year: We appreciate the reviewer’s suggestion, however, since the location and study year are clearly described in the Methods section, we believe repeating this information in each table title may be redundant. To maintain clarity and avoid unnecessary repetition, we kindly opted to keep the current table titles unchanged.

b) Remaining suggestions: All tables (Tables 1 through 4) were revised to incorporate the remaining suggestions.

---

## [Editor Report · Decision Letter 2]

Depression, anxiety and change in eating habits during the COVID-19 pandemic in Brazilian university students

PONE-D-24-00167R2

Dear Dr. Verly-Miguel,

We’re pleased to inform you that your manuscript has been judged scientifically suitable for publication and will be formally accepted for publication once it meets all outstanding technical requirements.

Kind regards,

Sandro Vieira Soares, Ph.D.

Academic Editor

PLOS ONE

---

## [Editor Report · Acceptance letter]

PONE-D-24-00167R2

PLOS ONE

Dear Dr. Verly-Miguel,

I'm pleased to inform you that your manuscript has been deemed suitable for publication in PLOS ONE. Congratulations! Your manuscript is now being handed over to our production team.

Kind regards,

on behalf of

Dr. Sandro Vieira Soares

Academic Editor

PLOS ONE